# Barriers to getting into postgraduate specialty training for junior Australian doctors: An interview-based study

**Belinda O'Sullivan**[1]*, **Matthew McGrail**[2], **Tiana Gurney**[1], **Priya Martin**[1]

**1** Rural Clinical School, The University of Queensland, Toowoomba, Queensland, Australia, **2** Rural Clinical School, The University of Queensland, Rockhampton, Queensland, Australia

* belinda.osullivan@uq.edu.au

## Abstract

### Background

Medical training is a long process that is not complete until doctors finish specialty training. Getting into specialty training is challenging because of strong competition for limited places, depending on doctors' chosen field. This may have a negative impact on doctor well-being and reduce the efficiency of the medical training system. This study explored the barriers of pre-registrar (junior) doctors getting into specialty training programs to inform tailored support and re-design of speciality entry systems.

### Methods

From March to October 2019, we conducted 32 semi-structured interviews with early-career doctors in Australia, who had chosen their specialty field, and were either seeking entry, currently undertaking or had recently completed various fields of specialty training. We sought reflections about barriers and major influences to getting into specialty training. In comparing and contrasting generated themes, different patterns emerged from doctors targeting traditionally non-competitive specialty fields like General Practice (GP) and other specialties (typically more competitive fields). As a result we explored the data in this dichotomy.

### Results

Doctors targeting entry to GP specialties had relatively seamless training entry and few specific barriers. In contrast, those pursuing other specialties, regardless of which ones, relayed multiple barriers of: Navigating an unpredictable and complex system with informal support/ guidance; Connecting to the right people/networks for relevant experience; Pro-actively planning and differentiating skills with recurrent failure of applications.

### Conclusions

Our exploratory study suggests that doctors wanting to get into non-GP specialty training may experience strong barriers, potentially over multiple years, with the capacity to threaten their morale and resilience. These could be addressed by a clearinghouse of information

**Data Availability Statement:** All relevant data are within the manuscript and its Supporting Information files.

**Funding:** The author(s) received no specific funding for this work.

**Competing interests:** The authors have declared that no competing interests exist.

about different speciality programs, broader selection criteria, feedback on applications and more formal guidance and professional supports. The absence of challenges identified for doctors seeking entry to GP could be used to promote increased uptake of GP careers.

## Introduction

Medical training is not complete until doctors gain postgraduate specialty qualifications. However, because there can be a large number of medical graduates competing for limited places on specialty programs, planning for and getting into specialty training may be challenging. Further, there are a wide range of specialties and sub-specialties, each with their own selection processes which may make it hard to target the right skills for eligibility. This may be compounded in countries where specialty selection occurs at least one year after exiting medical-school, thus pre-registrar doctors are employed in health services that may have a limited range of up-skilling opportunities to assist doctors to become more competitive for specialty applications. The process may have a negative impact on the well-being of early-career doctors and reduce the efficiency of the medical training system at producing qualified doctors. But despite these concerns, there is limited in-depth research exploring the barriers to getting into specialty training. Such evidence would be useful for developing more tailored support for early-career doctors and informing the design of postgraduate training systems for specialty entry.

This gap in the evidence is becoming more urgent for countries seeking to cater for an expanding pre-registrar doctor cohort [1,2]. This is fuelled by: 1) training more doctors [3]; 2) hospitals and health services recruiting overseas-trained doctors (whose qualifications may not be recognised) [4,5]; and 3) limited expansion of specialty training places to match demand. Beyond the number of places in any particular specialty programs, the complexity of entry criteria/process may relate to stronger barriers for getting into some specialty fields over others. In particular, more technical and narrow specialty fields may be more difficult to get into than broader specialties like general practice (GP) although this has never been explored.

The degree of pressure on individual doctors to manage the process of entering specialty training is an important issue to observe. In many countries, including the United Kingdom (UK), Europe, Australia and New Zealand, graduating medical students are initially required to work independently as pre-registrar doctors to gain enough experience to prepare for entry to individually governed specialty training programs [6,7]. Yet, it is a major transition to move from student to supervised intern, to independent (junior) doctor and then specialty registrar, covering diverse roles and responsibilities within different clinical teams and hospitals and health services [8]. Major national reviews recommend that these transitions should be managed to ensure that the expectations and needs of doctors are supported (to promote their satisfaction and sustainability), as well as to deliver the standards of care and balance of workforce needed by the community [7]. However, managing the step into specialty training is difficult without evidence about the barriers that pre-registrar doctors may experience.

Emerging UK research suggests that contemporary doctors are increasingly taking a break in practice before entering specialty training, for example in the pre-foundation year [9]. The reasons why have been scantily researched, suggesting this relates to training-work challenges at this stage such as exhaustion, stress and needing more time to prepare competitive portfolios for entering specialty training [10]. Australian research has also shown that pre-registrar doctors target research publications, likely as a means of building their portfolio of achievements

for getting into competitive specialties like surgery [11]. However, there has been limited qualitative research exploring this phenomenon and how it impacts doctors.

The broad theory about work psychology provides a useful lens upon which to evaluate the experiences of doctors in work and training environments, pursuing a goal of getting into specialty training [12,13]. Under the umbrella of this field is action theory which involves actions as goal-oriented behaviours aiming to achieve a career/work outcome [14]. It is concerned with the processes that intervene between the environment and the behaviour that may impact the achievement of goals such as getting into specialty training. Relative to a doctor's goal to enter specialty training, these processes include issues like orientation within a system (such as doctors collecting information), planning (such as listing sub-goals and back up plans, consciously or sub-consciously), executing the plan (such as taking advantage of opportunities, flexibility, speed and coordinating actions) and getting feedback (such as working out the level of progress towards the goal based on real-time or latent, verbal or non-verbal feedback) [14].

With this background and theoretical lens in mind, this study aimed to explore the barriers of pre-registrar doctors getting into postgraduate specialty training, to inform tailored support and re-design of speciality entry systems.

## Methods

### Setting

This study was based in Australia, which provides a useful case study of a country experiencing an abundance of hospital-based pre-registrar doctors after tripling medical school places, since 1996 (Box 1).

---

### Box 1: Context of study

In light of a burgeoning pre-registrar cohort, Australia is currently developing a new National Medical Workforce Strategy. It aims to achieve improved coordination of medical training from undergraduate to completion of postgraduate stages and reduce barriers to completing medical training, for a balanced generalist and specialist workforce, among other things [15]. After completing university-based medical school training, which is of 4–6 years duration, Australian doctors work independently as pre-registrars in hospitals, for a minimum of 2 years. Around this time, they are eligible to start applying for most specialties, which involves entering a competitive process for selection into one of a number of individually governed medical colleges (equivalent to 'residency' in many countries). Some specialties have additional requirements, which necessitate longer pre-registrar periods before selection is possible. In Australia, specialist training (typically spanning 3–6 years) is required for all specialties. After completing specialty training, doctors transition to a post-registrar phase as 'fellows' (of that specialty college), able to work independently with a specialist title. They usually earn better money at this point [16], and gain professional status and identity from a formal title and belonging to a tighter network [17].

---

Participants were sampled from an existing list of graduates of the University of Queensland (but no longer connected to the university) for whom we had contact details due to their links with a longitudinal workforce tracking project and agreed to be contacted for

participation in interviews [18]. These graduates were up to 17 years post-graduation which we thought a relevant group from whom to seek reflections about our research phenomenon as they were trying to get into, currently undertaking or recently completed specialty training. We targeted respondents of a mix of genders, work locations and specialties to get a wide range of data.

## Data collection

From March to October 2019, a semi-structured interview schedule informed by a topic guide (S1 File) explored doctors' reflections about their experiences at different career stages, including getting into specialty training. Doctors were encouraged to share openly from the perspective of their own experiences, in line with a phenomenological approach to qualitative research [19]. The interview schedule was piloted by mixed methods researchers with experience of studying medical education. Participants who responded were scheduled for video and phone interviews of up to 40 minutes with one of the PhD-trained qualitative researchers (TG and PM, both female academics). Interviewers had no prior relationship with participants. Participants gave informed consent and were not paid.

Interviewers met after each interview to discuss and share reflective notes about any emerging themes. On the basis of this, they refined prompts, which were used to expand understanding and enable thick description [20]. Interviews continued until saturation was achieved, as determined by the repetition of material in subsequent interviews and the research team not identifying any emerging areas that required further exploration in subsequent interviews. Interviews were recorded and transcribed verbatim.

## Analysis

The research team read the transcripts, in blocks of up to 9, using inductive coding for meaning, whereby emerging themes were strongly related to the data [21]. Related codes were grouped into clusters, thereby generating emerging themes. For the purpose of comparing and contrasting the data, annotations to quotations were used showing specialty field, participant age, gender, location of current work, so as to consider any interactions with the outcome. These annotations are defined in Table 1 During the analysis process, there were clear differences found between the experience of pre-registrars targeting (traditionally non-competitive fields) like General Practice (GP) and those pursuing non-GP specialties (typically more competitive fields to enter). As a result, we explored further data analysis along these lines.

The main author (BOS) led an in-depth review and analysis of clustered themes, iteratively checking the findings with the team for internal corroboration or disconfirmation until consensus was reached by the research team as to the final narrative that represented the data [22,23]. This process involved repeated reorganisation of the data as it was interpreted and re-interpreted through in-depth discussion by the research team, as well as through the lens of action theory and medical training literature. Moreover, the thematic labelling was refined over a 6-month period, to ensure it fit the observations in the data and was self-evident for policymaking. This reflexive process, involving continuously constructing and shifting understanding of the realities in the data (and challenging assumptions) by a diverse research team, helped to reduce subjective bias [20] and minimise any predilections or opinions [24,25].

The COnsolidated criteria on REporting Qualitative research (COREQ) checklist (S2 File) [26] and Tracy's qualitative research framework were used to guide qualitative methods [20]. This study had ethical approval from The University of Queensland human research ethics committee (#: 2012001171).

**Table 1. Definition of annotation used to depict textual data from interviews.**

| Item | Definition |
|---|---|
| *GP* [b] | General Practice |
| *Specialty training stage* [a] | |
| **Pre** | Pre-registrar, not yet training but identified targeted specialty, typically PGY1-5, already decided on specialty |
| Reg | Registrar currently undertaking postgraduate specialty training, typically PGY2-10 |
| Fell | Fellow, completed specialty training, in this study context, typically PGY 5–17 |
| *Work location* | |
| R | Rural (by Australian standards using the Modified Monash Model levels MMM 2–7) |
| M | Metropolitan |
| *Sex* | |
| Male | Male |
| Fem | Female |

[a] All participants interviewed had strong interest/plan or uptake of a specialty field allowing us to explore entry barriers.

[b] Non-GP specialties included in this study covered fields of: anaesthetics; ophthalmology; surgery; physician; radiology; psychiatry; oncology and; dermatology, as annotated in the quoted material.

# Results

Overall, 32 participants were included in our study until saturation was reached: 50% were female; 63% targeted non-GP specialties and were spread across as current pre-registrars (25%), registrars (31%) or recently fellowed as qualified specialists (44%). Reflections were fairly consistent across these groups as well as by age and location. But there was a strong split between the findings of GP and non-GP specialties.

The distinct difference noted between those targeting GP relayed was that their experience had minimal barriers, some noting that it: '*all just fell into place*' (GP_Fell_M6_Male) with '*getting places on the GP registrar program straight away*' (GP_Fell_M6_Male). They had strong career control:

> *I guess because I haven't been . . . trying to get into really competitive training programs, I've been able to shape what I wanted to do pretty well. (GP_Fell_R5_Fem)*

This finding was consistent despite our interviewers prompting reflection and discussion about this. After revisiting our analysis of transcripts, we confirmed an absence of their voice about barriers to entering GP training. Comparatively, for the group targeting other specialist fields, we identified three consistent barriers, each quite challenging. These are described below and summarised in Table 2.

### Navigating an unpredictable and complex system with informal support and minimal guidance

The first theme was that the number and complexity of non-GP specialty training options and the inconsistency of information about them made it '*very confusing*' to understand the process of getting into non-GP specialty training programs.

**Table 2. Key themes from the research, fitting for doctors targeting non-GP [a] specialties.**

| Theme title | Theme content |
|---|---|
| Navigating an unpredictable and complex system with informal support and minimal guidance | Many and complex of pathways<br>Volatile requirements<br>Rely on peer information about system/process<br>Minimal feedback<br>Perception of system bias/unwritten rules<br>Limited self-efficacy/ control |
| Connecting to enough of the right people and networks for relevant experience | Opportunistic connections<br>Being known to increase success<br>Relocation to major cities/ other states<br>Doing extra work to be known<br>Finding the 'right' people to impress |
| Pro-actively planning and differentiating skills but experiencing recurrent failure of applications | Planning early<br>Consciously building requisite CV<br>Differentiating skills<br>Facing choke points<br>Failing to be selected, morale, energy |

[a] non-GP specialties included respondents from: anaesthetics; ophthalmology; surgery; physician; radiology; psychiatry; oncology and; dermatology.

*. . .we all have found it very, very confusing as students because there's many different colleges, many different pathways, many different stages in training and it's not a consistent system throughout so it's hard to understand. (Psych_Reg_R3_Male)*

The entry requirements were considered volatile, and it was through hearsay that doctors learnt about what was needed to achieve success.

*. . .the ever-changing. . . requirements means that nobody ever actually quite knows what we need to do to get on to the program. . .I just heard things by hearsay and started ticking them off myself. (Orthopaedics_Pre_M4_Fem)*

With limited formal information about training program entry requirements, doctors heavily relied on sourcing this from trusted peers, just ahead on the career pathway.

*It's been helpful having a bunch of people around me who are on the same path, because we talk to each other and say, "This is what you need, you need to do these audits and you need to do these presentations and this is what the college likes." (Anaesthetics _Pre_R4_Fem)*

However, finding reliable peers to service the role of providing advice and mentoring was not always feasible, nor standardised. Further, the drive to stand out in some fields meant that this advice may not be sensitive.

*. . .while I had friends that were ahead of me in the training, they weren't people that I could necessarily go to with these sensitive information requests and advice and mentoring. (Surgery_Fell_M4_Male)*

The capacity for doctors to plan and benchmark their specialty training goals to address any gaps in experience was also hard based on the lack of feedback that non-GP colleges gave to applicants.

*. . . each year when you apply and you're not selected, there's minimal to no feedback whatso-ever. (Ophthalmology_Pre_M3_Male)*

This left doctors with limited feedback upon which to adjust their planning/execution of processes to achieve career goals, or to reorient these career goals. In particular, a lack of feedback left them questioning whether their efforts were targeted enough, or college selection was biased.

*A mate, I think it took him eight consecutive years of applying before he got on to orthopaedic surgery. That seems like a huge waste of time. . . was it just because he just had to get his CV up to scratch or was it because of systems that are biased? (Surgery Fell_M4_Male)*

*. . .many of them [Colleges] don't have, shall we say, transparent selection criteria. So, you'll find lots of very gifted or very able applicants aren't selected for particular reasons. (Ophthalmology_Pre_M3_Male)*

## Connecting to enough of the right people and networks for relevant experience

The second theme around barriers were that getting into non-GP specialty training depended on developing connections to the 'right people' considered critical for getting relevant experience that would increase the chances of getting into non-GP training programs. This could start early in medical training.

*. . . meeting him [a dermatologist during medical school] was also pivotal in terms of building my CV to be eligible. (Dermatology_Reg_M4_Male)*

Further, being aligned and known within clinical networks was sought after to assist doctors to access suitable jobs and get accepted into non-GP specialty training.

*Working for five years in the same hospital I'm very well-known. . .it's easy to go to a department and say, "You know me and you know what I've done and you know that I'm good because we've worked together, and so give me a job". (Anaesthetics_Pre_R4_Fem)*

*. . .it certainly would be easier to get on to the training pathway [in city x] here as opposed to back in [city y] where no-one really knows us. (Ophthalmology_Pre_M5_Fem)*

Relocating was commonly required to gain specific experience so as to improve the competitive edge for non-GP specialty entry.

*Unfortunately, in [state x] . . .There's not much in the way of job opportunities or research projects in order to develop your CV [curriculum vitae]. So, that's what drove me to relocate to [city x]. . .in the five years in which I've been practising, I've relocated from [city x] to [region x] to [city y] to [city z] and then back to [city y] again. (Ophthalmology_Pre_M3_Male)*

The sub-specialist experiences sought after were largely in major city hospitals.

*I moved to [hospital x] in [city x] basically because of opportunities and exposure to sub-specialties which aren't available at the peripheral centres. (Physician_Reg_M2_Male)*

Some reported that undertaking research was a way to build relevant connections for the right jobs.

*. . .their name gets known. They're getting published. They've made connections so that they can get these jobs that they want. (GP_Pre_M6_Fem)*

Once again, finding the right people and networks to know relied on guidance from peers.

*. . .you just learn from talking to other people who are in the year above you or whatever, that "These are the people you need to go and talk to and try and impress to have a chance of getting this job. . .." (Physician_Reg_M2_Male)*

## Pro-actively planning and differentiating skills but experiencing recurrent failure of applications

Those targeting non-GP specialties used a highly methodical and conscious approach to building the requisite curriculum vitae for eligibility to non-GP specialty training.

*[When in medical school]. . .I made the very conscious effort of identifying which competitive specialties I would be interested in and. . .then doing things to build my CV [curriculum vitae] towards it. (Dermatology_Reg_M4_Male)*

Differentiating themselves added large volumes of additional work, particularly as doctors sought wider exposure and across multiple non-clinical domains of work.

*I have worked incredibly hard for the last two years to make my résumé what it is. So, I've done two audits, six courses. I've given eight presentations, I've presented to the state-wide level. (*achievements_Pre_R4_Fem)

Despite their planning, doctors had a sense of limited control over outcomes, relaying that there were many 'barriers', 'hurdles' and 'choke points' related to getting into non-GP specialty training.

*. . .There's a million different choke points. (Surgery_Fell_M4_Male)*

One also noted that the need to regularly and concurrently apply for jobs and training programs impacted morale.

*. . .so I've applied for the training program twice and not gotten on each time. . . But also, having to apply for unaccredited jobs year after year and not getting selected. . .that's also very difficult in terms of your morale. (Ophthalmology_Pre_M3_Male)*

If applications for training failed, some doctors felt trapped, particularly if their skills had become very focused over the years of preparing to enter training in a particular field. Meanwhile they realised that they were facing more competition as new cohorts of medical graduates entered the pre-registrar pool each year.

*I think it's difficult being a PGY-5 because, so far, this is my third year of doing orthopaedics exclusively and it's a bit pigeon-holing. (Orthopaedics_Pre_M4_Fem)*

For some, with set ideation to train in a non-GP specialty, contemplating not getting into the training in their area of preference made them reflect on whether they would continue with a career in medicine.

> . . .if I wasn't successful on getting on to anaesthetics I don't really think there's another field of medicine that I would want to work in and I would potentially think of leaving medicine . . .Medicine is not compatible with happiness. (Anaesthetics_Pre_R4_Fem)

## Discussion

This exploratory research provides a unique in-depth analysis of the barriers related to doctors getting into specialty training, in the context of Australia's medical training system. Our research suggests there may be distinct differences between pre-registrars targeting GP or non-GP specialties. The former group had a relatively seamless entry to specialty training and good career control, potentially as the most available places exist in GP training of any specialty field. For those pursuing non-GP specialties, they faced multiple challenges that resonated across various fields, including surgery and internal medicine. Through the lens of work action theory [14], pre-registrar doctors aiming for non-GP fields sought to orientate themselves to the specialty training systems relevant to themselves, but they faced a lack of clear information about the range of non-GP training options. Instead, they noted complex and changing information about requirements, mostly sourced through peers. Although pre-registrar doctors undertook early and active planning and executed these plans purposefully and methodically to achieve entry to non-GP specialties, this entailed additional non-clinical workload and relocation requirements. Some doctors may find these requirements hard to meet depending on social and economic cost, including disrupting a connection to place, for doctors engaged and interested in staying attached to rural medicine.

Despite their planning and action, achieving entry to non-GP specialty training commonly failed, suggesting doctors had poor control over this goal. Whilst doctors might seek to adjust career goals when plans fail, our research suggests doctors targeting non-GP specialties may receive limited feedback from the specialty/job application process. This may make it hard to re-engage with the desired goal through planning and executing new strategies as well as difficult to refine original goals towards realistic outcomes and timeframes. Further, the ability to change their specialty focus was difficult as they had pursued experience in one focused area as a pre-condition of achieving entry to training in a non-GP specialty, possibly at the expense of other areas [14]. Our findings reinforce those of other studies showing that pre-registrar doctors may have limited occupational control [27], although previous research has not yet explored this specifically in relation to getting into various specialty training. Otherwise, there are few other studies about the experience and challenges to accessing specialty training amongst junior doctors already in the workforce upon which to contrast our findings.

The implications are that these barriers may intersect over many years for pre-registrar doctors who are not successful at getting into specialty training, potentially diminishing their morale and resilience. This may require that hospitals and health services embrace training-related pressures as part of their well-being policies for junior doctors. The barriers we found may divert some pre-registrar doctors to poorly suited career options, with implications of wasting talent, and causing dissatisfying careers and potentially reducing medical service productivity. Further, while doctors remain in the hospital system as they prepare to enter specialty training, they use more public funding than when they qualify and move into mixed or private practice roles.

A potential part of the solution is developing a national clearinghouse of up-to-date information about non-GP specialty training parameters. This differs from the current situation where each college hosts this information, with varying degrees of clarity and updating. Information managed in one place would help early-career doctors to compare program requirements. If the criteria to enter specialty programs were also broader, they may overlap more between different fields, allowing early-career doctors to enjoy more diverse early career experience, and being able to change career directions without major implications.

Another avenue to improve system design includes increasing formal career guidance and professional supports for doctors throughout the first 2–5 years of practise [8]. Critically this advice should reduce the need to source information from peers a few years ahead, which is not necessarily accessible, sensitive, or objective in nature. Finally, hospital and health services could reduce the need for pre-registrars to relocate if they engaged more local up-skilling and networking with specialist clinical groups for emerging doctors with clinical networks.

Our findings of a relatively seamless pathway for doctors to get into GP specialties, could be used as a valuable marketing tool to promote more contemporary doctors to pursue careers in General Practice. This could have a strong community benefit for countries like Australia that are experiencing overspecialisation of their medical workforce [15,28].

Our study is exploratory and small given how many specialty fields we included, and we suggest more research could expand on different barriers by field. Although most barriers centred on the 63% of respondents targeting non-GP specialties, the findings about General Practice were pertinent and ours is the only study to do this comparative work across diverse specialties to tease this out. Using interviews allowed us to collect in-depth material about the complexities of barriers, building on other survey-based research, but the data may not be representative. Although our participants were from a single university cohort, they were working independently of the university, in diverse settings and fields, allowing for rich exploration. Respondents covering a range of early-career stages and fields provided a broad aggregate perspective of the key barriers. The results should be interpreted with caution in other contexts where postgraduate medical education policies and practice context differ.

## Conclusions

In conclusion, many pre-registrar doctors pursuing entry to training in non-GP specialty fields face major intersecting challenges. These include a lack of formal information about complex pathways, pressure to connect to people and get enough experience, and repeatedly failing despite trying to differentiate themselves. These challenges may be chronic, or occur at different times over multiple years, placing pressure on their morale and resilience. The challenges could be addressed by: a single clearinghouse of information about speciality training programs, broader specialty selection criteria, feedback on applications and more formal guidance and professional supports. We identified seamless entry and strong career control for doctors getting into training in GP specialties which could be leveraged to promote the uptake of careers like General Practice.

## Supporting information

**S1 File. Interview guide.**
(DOCX)

**S2 File. COREQ checklist.**
(PDF)

## Acknowledgments

We acknowledge the participation of the 32 doctors who were interviewed for this study.

## Author Contributions

**Conceptualization:** Belinda O'Sullivan, Matthew McGrail.

**Data curation:** Belinda O'Sullivan, Tiana Gurney.

**Formal analysis:** Belinda O'Sullivan, Matthew McGrail, Tiana Gurney, Priya Martin.

**Investigation:** Tiana Gurney, Priya Martin.

**Methodology:** Belinda O'Sullivan.

**Supervision:** Matthew McGrail.

**Writing – original draft:** Belinda O'Sullivan.

**Writing – review & editing:** Belinda O'Sullivan, Matthew McGrail, Tiana Gurney, Priya Martin.

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
