## [Decision Letter · Decision Letter 0]

4 Jun 2021

PONE-D-21-08376

Barriers to getting into postgraduate specialty training for junior doctors: an interview-based study

PLOS ONE

Dear Dr. McGrail,

Thank you for submitting your manuscript to PLOS ONE. After careful consideration, we feel that it has merit but does not fully meet PLOS ONE’s publication criteria as it currently stands. Therefore, we invite you to submit a revised version of the manuscript that addresses the points raised during the review process.

Please address reviewer's comments particularly how findings related to theory.

We look forward to receiving your revised manuscript.

Kind regards,

Webster Mavhu

Academic Editor

PLOS ONE

Journal Requirements:

3. Please provide additional details regarding participant consent. In the Methods section, please ensure that you have specified (1) whether consent was informed and (2) what type you obtained (for instance, written or verbal). If your study included minors, state whether you obtained consent from parents or guardians. If the need for consent was waived by the ethics committee, please include this information.

Additional Editor Comments:

Please address reviewer's comments including how findings link to theory.

Interview guide should be supplied as supporting information (S1 file) and only referenced in body of manuscript.

Reviewers' comments:

Reviewer's Responses to Questions

**Comments to the Author**

1. Is the manuscript technically sound, and do the data support the conclusions?

Reviewer #1: Partly

2. Has the statistical analysis been performed appropriately and rigorously? 

Reviewer #1: N/A

3. Have the authors made all data underlying the findings in their manuscript fully available?

Reviewer #1: Yes

4. Is the manuscript presented in an intelligible fashion and written in standard English?

Reviewer #1: Yes

5. Review Comments to the Author

Reviewer #1: I have attached a word document with my full response.

Many thanks for the opportunity to review this manuscript, the methodology is sound and the conclusions in the most part seem reasonable. However, I think they could be clearer at times. I also am not sure about their conclusion regarding GP training. In addition I wonder if more explicitly linking their findings to the theory that underpins this may help make the paper clearer, and recommendations easier to implement.

6. PLOS authors have the option to publish the peer review history of their article (what does this mean?). If published, this will include your full peer review and any attached files.

Reviewer #1: **Yes: **Dr Marianne McCallum

---

## [Author Response · Author response to Decision Letter 0]

20 Jun 2021

Dear Editor of PlosONE

Reference: PONE-D-21-08376.

Thank you for recently peer reviewing our article ‘Barriers to getting into postgraduate specialty training for junior doctors: an interview-based study’.

We have responded to each of the reviewer’s comments by attaching a table where we noted changes to the manuscript and included a tools tracked and clean version of the manuscript. We believe that these changes have substantially strengthened the paper.

We thank you for considering our manuscript and we look forward to hearing from you.

Kind regards

Dr Belinda O’Sullivan (Corresponding Author)

---

## [Editor Report · Decision Letter 1]

7 Jul 2021

PONE-D-21-08376R1

Barriers to getting into postgraduate specialty training for junior doctors: an interview-based study

PLOS ONE

Dear Dr. O'Sullivan,

Thank you for submitting your manuscript to PLOS ONE. After careful consideration, we feel that it has merit but does not fully meet PLOS ONE’s publication criteria as it currently stands. Therefore, we invite you to submit a revised version of the manuscript that addresses the points raised during the review process.

A few comments/suggestions are attached.

We look forward to receiving your revised manuscript.

Kind regards,

Webster Mavhu

Academic Editor

PLOS ONE

Journal Requirements:

Additional Editor Comments (if provided):

A few minor comments are in attached document.

---

## [Author Response · Author response to Decision Letter 1]

7 Jul 2021

See attached response to reviewers

---

## [Editor Report · Decision Letter 2]

9 Jul 2021

PONE-D-21-08376R2

Barriers to getting into postgraduate specialty training for junior Australian doctors: an interview-based study

PLOS ONE

Dear Dr. O'Sullivan,

We are about to accept the manuscript. However, there are a few suggestions/edits that need your attention before we do so - see attached document.

We look forward to receiving your revised manuscript.

Kind regards,

Webster Mavhu

Academic Editor

PLOS ONE

Journal Requirements:

Additional Editor Comments (if provided):

Please ensure title is consistent - Title page says ...junior Australian doctors...Line 16 says ...Australian junior doctors..

See other minor suggestions/edits in attached.

---

## [Author Response · Author response to Decision Letter 2]

15 Jul 2021

Please note that we have updated ethics to state that ‘Participants gave verbal informed consent at the beginning of the interview’ and have updated the Ethics Statement field of the submission form to reflect this.

---

## [Editor Report · Decision Letter 3]

21 Jul 2021

PONE-D-21-08376R3

Barriers to getting into postgraduate specialty training for Australian junior doctors: an interview-based study

PLOS ONE

Dear Dr. O'Sullivan,

There are a few more edits/suggestions in attached document.

We look forward to receiving your revised manuscript.

Kind regards,

Webster Mavhu

Academic Editor

PLOS ONE

Journal Requirements:

Additional Editor Comments:

There are a few suggestions/edits in attached - see for example line 134.
---

## [Author Response · Author response to Decision Letter 3]

25 Jul 2021

Thank you for recently editing our article ‘Barriers to getting into postgraduate specialty training for Australian junior doctors: an interview-based study’. We have uploaded a response to editor document outlining the amendments.

We thank you for working with our manuscript and we look forward to hearing from you.

---

## [Editor Report · Decision Letter 4]

1 Oct 2021

Barriers to getting into postgraduate specialty training for Australian junior doctors: an interview-based study

PONE-D-21-08376R4

Dear Dr. O'Sullivan,

We’re pleased to inform you that your manuscript has been judged scientifically suitable for publication and will be formally accepted for publication once it meets all outstanding technical requirements.

Kind regards,

Webster Mavhu

Academic Editor

PLOS ONE
---

## [Editor Report · Acceptance letter]

12 Oct 2021

PONE-D-21-08376R4 

Barriers to getting into postgraduate specialty training for junior Australian doctors: an interview-based study 

Dear Dr. O'Sullivan:

I'm pleased to inform you that your manuscript has been deemed suitable for publication in PLOS ONE. Congratulations! Your manuscript is now with our production department. 

Kind regards, 

on behalf of

Dr. Webster Mavhu 

Academic Editor

PLOS ONE